# Nanomedicine for Combination Urologic Cancer Immunotherapy

**DOI:** 10.3390/pharmaceutics15020546

**Published:** 2023-02-06

**Authors:** Yun Tian, Zhenzhu Liu, Jianbo Wang, Linan Li, Fuli Wang, Zheng Zhu, Xuejian Wang

**Affiliations:** 1Department of Urology, The First Affiliated Hospital of Dalian Medical University, Dalian 116021, China; 2Department of Cardiovascular, The Second Affiliated Hospital of Dalian Medical University, Dalian 116023, China; 3Department of Orthopedics, The First Affiliated Hospital of Dalian Medical University, Dalian 116011, China; 4Department of Urology, Xijing Hospital, Fourth Military Medical University, Xi’an 710032, China; 5Department of Medicine, Brigham and Women’s Hospital, Harvard Medical School, Boston, MA 02115, USA

**Keywords:** nanomedicine, immunotherapy, urologic cancer, immune checkpoint inhibitors

## Abstract

Urologic cancers, particularly kidney, bladder, and prostate cancer, have a growing incidence and account for about a million annual deaths worldwide. Treatments, including surgery, chemotherapy, radiotherapy, hormone therapy, and immunotherapy are the main therapeutic options in urologic cancers. Immunotherapy is now a clinical reality with marked success in solid tumors. Immunological checkpoint blockade, non-specific activation of the immune system, adoptive cell therapy, and tumor vaccine are the main modalities of immunotherapy. Immunotherapy has long been used to treat urologic cancers; however, dose-limiting toxicities and low response rates remain major challenges in the clinic. Herein, nanomaterial-based platforms are utilized as the “savior”. The combination of nanotechnology with immunotherapy can achieve precision medicine, enhance efficacy, and reduce toxicities. In this review, we highlight the principles of cancer immunotherapy in urology. Meanwhile, we summarize the nano-immune technology and platforms currently used for urologic cancer treatment. The ultimate goal is to help in the rational design of strategies for nanomedicine-based immunotherapy in urologic cancer.

## 1. Introduction

The incidence and mortality of urologic cancers, including renal cell carcinoma, uroepithelial carcinoma, and prostate cancer, have remained high in recent years. In addition, the overall incidence of urologic cancers, particularly the overall incidence of prostate cancer, has been higher in developed regions than in developing regions. It is estimated that there will be 268,490 new cases and 34,500 deaths from prostate cancer in the United States only in 2022 [1]. Although surgery, conventional chemotherapy, and radiotherapy can improve the outcome, the therapeutic response is still unsatisfactory. Therefore, improving treatment efficacy and early diagnosis is pivotal.

Tumor immunotherapy is a novel therapeutic approach to enhance the immune response against malignant tumors. The human immune system suppresses tumor growth and metastasis; however, tumor cells evade the immune system by establishing an immunosuppressive tumor microenvironment. T cell failure or dysfunction is now believed to be the key mechanism leading to the impaired immune response to tumors. A major hallmark of T cell failure is the enhanced expression of several immune checkpoints, such as programmed death-1 (PD-1) and cytotoxic T lymphocyte antigen-4 (CTLA-4). Consistently, it has been experimentally demonstrated that T cell function decreases with increased expression of immune checkpoints [2]. PD-1 is a transmembrane protein widely expressed by T cells, B cells, NK cells, and dendritic cells. The interaction between PD-1 and its ligand, programmed death-ligand 1 (PD-L1), can inhibit T cell activation and suppress the immune response [3]. CTLA-4 is present on the surface of T cells. It is structurally similar to CD28 and binds CD80 and CD86 with a higher affinity than that of CD28, thereby inhibiting T cell activation [4]. Therefore, blocking PD-1 or CTLA-4 can restore T cell function or viability and increase the anti-tumor immune response. The first immune checkpoint inhibitor, a CTLA-4 inhibitor (ipilimumab), received FDA approval in 2011 for treating advanced melanoma. Thereafter, other immune checkpoint inhibitors, PD-1 inhibitors (pembrolizumab and nivolumab), were approved. Currently, immune checkpoint inhibitors are recognized and recommended by the European Association of Urology (EAU), the European Society of Medical Oncology (ESMO), and the National Comprehensive Cancer Network (NCCN) for the treatment of urologic tumors [5,6,7]. Nivolumab has been used as a second-line treatment for renal cell carcinoma and bladder cancer. In a phase II clinical trial (KEYNOTE-052) (NCT02335424), pembrolizumab showed anti-tumor activity and tolerability in patients unsuitable for cisplatin chemotherapy (NCT02335424) [8]. As research continues to progress, more immunotherapy modalities are being introduced to clinical trials, such as non-specific activation of the immune system by cytokines and BCG, pericyte immunotherapy, and tumor vaccines [9]. Although immunotherapy can improve the outcome of patients with advanced urologic tumors, it still faces challenges such as systemic immune-related adverse events (irAEs), low response rates, and poor targeting due to the complex microenvironment of tumors.

Recently, nanotechnology has rapidly developed in biomedicine and increased the efficacy of cancer treatment. Nanomedicine can precisely target tissues or cells to enhance the immune response and decrease irAEs [10]. Nanodrugs can carry two or more drugs to improve synergistic effects [11]. They can also modulate the pharmacokinetics of drugs by adjusting their size and surface properties [12]. These advantages made nanoparticles ideal candidates for the diagnosis and treatment of cancer. Clinical trials have demonstrated the feasibility and tolerability of thermal therapy by magnetic nanoparticles for locally recurrent prostate cancer [13]. In addition, iron uptake by tumor cells was observed using iron oxide nanoparticles in tumor-bearing mice [14]. The combination of nanotechnology with other therapeutic strategies, including ICIs, vaccines, and imaging, facilitated cancer treatment.

In this article, we present current immunotherapy modalities, including immune checkpoint blockade, non-specific activation of the immune system, adoptive cell therapy, and tumor vaccine. Particularly, we focus on those immunotherapy modalities that can achieve clinical translation. Then, we summarize the existing methods of nanotechnology for the treatment and diagnosis of urologic tumors and attempt to address the shortcomings of conventional ICIs, vaccines, magnetic resonance imaging (MRI), and computed tomography (CT).

## 2. Renal Cell Carcinoma

Renal cell carcinoma (RCC) is the most common type of kidney cancer and has several subtypes with distinct features, and clear cell carcinoma (ccRCC) is the most common. RCC has long been thought to respond to immunotherapy; therefore, several immunotherapy modalities, such as immune checkpoint inhibitors and vaccines, have been approved for RCC.

### 2.1. Immunotherapy of Renal Cell Carcinoma

PD1 inhibitors have been approved for the treatment of RCC and are now the new first-line standard of care for patients with intermediate-risk and low-risk metastases. Furthermore, monotherapy with nivolumab, a PD1 inhibitor, is the standard second- or third-line treatment after vascular endothelial growth factor (VEGF) and tyrosine kinase inhibitors (TKIs) [15,16]. Nivolumab is a fully humanized immunoglobulin G4 programmed death-1 immune checkpoint antibody. It restored T cell immune function and improved objective response rates (ORR) of patients with metastatic RCC (mRCC) in a phase II clinical trial (NCT01354431) [17]. In a subsequent phase III trial with 44 patients with advanced non-ccRCC who were treated with nivolumab, an ORR of 13.6% (95% confidence interval (CI), 5.2–27.4) and a median OS of 13.6 months (95% CI, 9.2–not estimable) were achieved (NCT02596035) [18]. Another phase III trial was conducted with 821 patients with advanced ccRCC who were treated with one or two anti-angiogenic regimens and received the biweekly injection of nivolumab or daily administration of an oral mTOR inhibitor, everolimus. The study reported a median OS of 25.0 months in the nivolumab group (95% CI, 21.8 to not estimable) and an objective remission rate superior to everolimus (25% vs. 5%; *p <* 0.001). Median progression-free survival was similar between groups (4.6 months vs. 4.4 months; HR 0.88; *p* = 0.11), but fewer grade 3 or 4 adverse events occurred in the nivolumab group compared to the everolimus group (NCT01668784) [16]. These findings led to the approval of nivolumab for metastatic RCC in November 2015. In 2018, a phase III trial consisting of 1096 patients with untreated advanced ccRCC was conducted to compare intravenous nivolumab plus Ipilimumab four times every three weeks with oral sunitinib once daily for four weeks. The 18-month OS rate was 75% (95% CI, 70–78) for nivolumab plus Ipilimumab and 60% (95% CI, 55–65) for sunitinib over a median follow-up of 25.2 months in low- and intermediate-risk patients. Median OS was not achieved for nivolumab plus Ipilimumab vs. 26.0 months for sunitinib (hazard ratio (HR) for death, 0.63; *p <* 0.001). The objective remission rate was 42% vs. 27% (*p <* 0.001), and the complete remission rate was 9% vs. 1%. Median progression-free survival was 11.6 months and 8.4 months, respectively. Treatment-related adverse events occurred in 93% of the nivolumab plus Ipilimumab group compared with 97% in the sunitinib group (NCT02231749) [19]. Based on the results of this trial, nivolumab and Ipilimumab combination therapy was approved as the first-line treatment for patients with low- to intermediate-risk advanced RCC.

PD-L1 inhibitors, such as atezolizumab, durvalumab, and avelumab, have been used, mostly in combination, to treat RCC in clinical trials. Again, PD-L1 inhibitors showed promising therapeutic results and safety. Artarone et al. conducted a meta-analysis to confirm the superiority of ICIs over chemotherapeutic agents in pretreated non-small-cell lung cancer (NSCLC) patients and showed a slight benefit of anti-PD-1 inhibitors (nivolumab and pembrolizumab) over anti-pd-l1 (atezolizumab) inhibitors [20]. In addition, a meta-analysis by Petrelli et al. demonstrated significantly improved outcomes and survival with ICI combined chemotherapy treatment compared to platinum-based combination chemotherapy in untreated NSCLC patients [21]. These experiments will provide some new ideas and confidence for the application of chemotherapy combined with immunotherapy in urological tumors. In a phase 1 pilot study, 70 patients with metastatic RCC, including ccRCC and non-ccRCC, were treated with intravenous atezolizumab every 3 weeks. The study reported an OS of 28.9 months (median, 95% CI, 20.0 months to not reached), progression-free survival of 5.6 months (median; 95% CI, 3.9 to 8.2 months), and an objective remission rate of 15% (95% CI, 7% to 26%). Grade 3 treatment-related and irAEs occurred in 17% and 4% of patients, respectively, and no grade 4 or 5 adverse events were reported [22]. These findings demonstrated that atezolizumab has a promising efficacy and manageable safety for treating RCC. In another trial, compared with sunitinib, the combination of atezolizumab and bevacizumab prolonged progression-free survival with a favorable safety profile in patients with metastatic RCC (NCT02420821) [23]. Similarly, durvalumab was used in combination with MEDI0680. MEDI0680, also called AMP-514, is a humanized immunoglobulin *γ*-4 kappa (IgG4k), which is specific for PD-1 protein. Compared with durvalumab monotherapy, the combination of MEDI0680 and durvalumab was safe and tolerable but did not improve the outcome of advanced ccRCC (NCT02118337) [24]. In addition, the safety and efficacy of avelumab in combination with axitinib are being evaluated in untreated advanced RCC (NCT02493751). Furthermore, the efficacy of avelumab in combination with axitinib vs. sunitinib is being tested as the first-line therapy in patients with advanced RCC (NCT02684006).

Ipilimumab is an anti-CTLA-4 IgG1 monoclonal antibody that was first introduced as a first-line treatment option for intermediate-to-advanced or metastatic melanoma [25]. As mentioned previously, Ipilimumab is currently used in combination with nivolumab for the treatment of RCC. Moreover, another CTLA-4 inhibitor, tremelimumab, in combination with durvalumab, is being tested in various cancers, such as breast cancer, ovarian cancer, colorectal cancer, cervical cancer, and RCC (NCT01975831).

### 2.2. Adoptive Cell Immunotherapy (ACI)

Since 1990, different methods of ACI, using lymphokine-activated killer cells (LAK cells) [26,27], tumor-infiltrating lymphocytes (TILs) [28,29], and cytokine-induced killer cells (CIK), have been assessed. Compared with other immunotherapies, the remission and survival rates achieved by ACI were inconsistent in different studies. In a phase III trial, treatment with CD8^+^ TILs did not improve the response rate or survival in patients treated with low-dose recombinant interleukin-2 (rIL-2) after nephrectomy [30]. CIK immunotherapy improved the prognosis of metastatic ccRCC in a randomized study, and more cycles of CIK therapy was associated with a better outcome [31]. However, CIK combination therapy for NSCLC has been extensively studied, with encouraging results [32,33], and in a phase IB trial, autologous CIK cell immunotherapy in combination with Sintilimab plus chemotherapy was well-tolerated and showed encouraging results in previously untreated patients with advanced NSCLC (NCT03987867) [34]. Currently, chimeric antigen receptor (CAR)-T cell therapy is attracting widespread attention. Ji et al. enhanced CD70 CAR-T cell-mediated RCC regression by activating the cGAS-STING pathway, suggesting a promising adjuvant therapeutic strategy for CAR-T cell therapy in solid tumors [35].

### 2.3. Vaccines

The vaccines are currently primarily used to treat RCC, not to prevent it. AGS-003, a dendritic cell-based vaccine, was used in a phase II trial with 21 intermediate- and low-risk patients who received sunitinib in the first week, followed by 5 doses of AGS-003 every 3 weeks and then every 12 weeks. The study reported a median PFS of 11.2 months (95% CI 6.0, 19.4) and a median OS of 30.2 months (95% CI 9.4, 57.1). In addition, 7 (33%) patients survived at least 4.5 years and 5 (24%) survived more than 5 years (NCT00678119) [36]. In a phase III trial, an AGS-003 plus sunitinib regimen for metastatic RCC was discontinued due to the lack of efficacy [37]. IMA901 is the first vaccine for RCC, consisting of multiple tumor-associated peptides (TUMAPs). In a phase III study with 339 patients with metastatic RCC who received sunitinib plus IMA901 or sunitinib alone, the median OS was 33.17 months (95% CI 27.81–41.36) in the sunitinib plus IMA901 group and was not achieved (33.67 to not achieved) in the sunitinib monotherapy group, with no statistical significance. Additionally, 116 (57%) cases in the sunitinib plus IMA901 group and 62 (47%) cases in the sunitinib group experienced grade 3 or more serious adverse events. These findings showed that the addition of IMA901 to sunitinib did not improve OS [38]. Another DC vaccine, AdGMCA9, showed a better safety profile in phase 1 clinical trials of metastatic RCC, without causing any serious adverse events (NCT01826877) [39].

However, these methods of immunotherapy still face many challenges. For example, the drugs do not specifically accumulate in tumor cells; thus, triggering unwanted systemic reactions, a shorter half-life, more severe irAEs, and many other problems.

### 2.4. Nanoparticles in Renal Cell Carcinoma

The rapid development of nanotechnology in recent years has provided new strategies for immunotherapy and diagnosis of urologic tumors. Nanoparticles possess small sizes (1–1000 nm), high reactivity, and a delivery effect [40]. Nanoparticles can be actively or passively targeted (enhanced permeability and retention (EPR)) and cross the biological barrier to reach the specific sites or cells. Nanoparticles can be applied to recognize not only tumor cells, but also matrix metalloproteinases (MMPs) or lysyloxidase (LOX) in the tumor microenvironment (TME), thus realizing dual-recognition. Furthermore, the use of nanoparticles is able to induce immunogenic cell death (ICD) and release danger-associated molecular patterns (DAMPs) for the purpose of promoting dendritic cells’ (DCs) maturation, whereas nanoparticles can also convert M2-type macrophages into M1-type macrophages in the TME. The role of nanoparticles in urologic tumors is summarized in Figure 1.

The unique characteristic of urological tumor cells is that tumor cells are located between the tumor microenvironment and the urinary tract. Nanoparticles can carry two or more fluorescent probes at the same time, which can specifically identify tumor cells and the overexpression of the protease family (LOX, MMPs, etc.) around the tumor, so as to achieve multiple recognition. Thus, tumors can be detected through blood/urine samples and external imaging systems. Meanwhile, nanoparticles can carry drugs and directly kill tumors after they are identified. In addition, nanoparticles can directly cause an immune response in the tumor microenvironment. Nanoparticles can convert M2-type cells into M1-type cells inside tumors, improve immunogenic cell death (ICD), and promote DC cell maturation, all of which improve anti-tumor immunity. To improve tumor therapy, nanoparticles can trigger ferroptosis while also increasing anti-tumor immunity in solid tumors. 

#### 2.4.1. Diagnosis

Currently, most of the screening instruments, such as MRI, CT, and positron emission tomography (PET), can detect tumors only when significant changes have occurred. How-ever, tumor cells have already proliferated or metastasized by this stage. This issue can be solved by nanotechnology [41]. Nanoparticles can be used as contrast agents for CT and MRI [42]. In addition, nanomaterials can be used for more sensitive and accurate detection of biomarkers [43], leading to earlier and more precise diagnosis of cancers.

AS1411 is a guanine-rich oligonucleotide (GRO), which recognizes and internalizes nucleosides in various cancer cells. It is the first aptamer used in clinical trials of cancer. Zheng et al. used Mn-MoS2 quantum dots to modify the AS1411 aptamer to develop a promising fluorescent/MR dual-modality imaging probe for the accurate diagnosis of RCC [44]. AS1411-Mn-MoS2 quantum dots not only exhibited low toxicity in RCC-bearing mice, but also showed specific MRI signal enhancement. For the first time, Ordikhani et al. designed a nanocarrier, which selectively interacted with proximal tubular epithelial cells (PTECs) and RCC in the kidney. PEGylated polylactic-co-glycolic acid (PLGA) nanoparticles of lambda light chains (LCs) improved the diagnosis of RCC in tumor-bearing mice by interacting with membrane protein macro-proteins [45]. Using gold nanoparticle-based enhanced targeting (AuNPET), Arendowski et al. identified four potential biomarkers in the serum and urine of patients with confirmed RCC for differentiating between patients with ccRCC and healthy controls. They also found four biomarkers for differentiating between ccRCC with and without metastasis, six biomarkers for differentiating between low (T1 and T2) and high (T3 and T4) stages, and six biomarkers for differentiating between low (Fuhrman I and II) and high (Fuhrman III and IV) grades of ccRCC. These biomarkers can be used to identify the type, grade, and stage of RCC [46]. Guimaraes et al. used magnetic nanoparticles (MNP) of lymphotropic nanoparticle-enhanced MRI (LNMRI) to detect metastatic LN in RCC patients with high sensitivity (100%) and specificity (95.7%) [47]. Lu et al. used superparamagnetic iron oxide (SPIO) nanoparticles and mAb G250 as molecular magnetic resonance imaging (mMRI) probes. mAb G250 specifically detected carbonic anhydrase IX (CAIX) antigen in ccRCC, and SPIO nanoparticles as an MRI contrast agent presented an excellent MRI response. In vitro studies showed that the mAb G250-SPIO nanoprobe can be successfully used for specific labeling of ccRCC cells [48].

#### 2.4.2. Treatment

The multifunctional properties of nanoparticles make them have great advantages in tumor immunotherapy. Nanoparticles can not only provide a controlled thermal dose and targeted therapy for hyperthermia or PDT to enhance ICD, but also combine drugs with multiple properties that can be co-delivered to the tumor sites. Moreover, multiple ligands can be arranged on the surfaces of polymers and nanoparticles to facilitate the binding of immunostimulatory receptors. The primary limitation of immunotherapy is the high level of toxicity observed when administered systemically and stimulating circulating lymphocytes. Immunotherapeutic agents coupled with nanoparticles can be delivered to tumors in a much shorter time while being cleared from the body circulation more quickly, providing similar stimulation in the TME, but significantly reducing systemic exposure.

Excessive and uncontrolled neovascularization leads to vascular leakage and poor lymphatic drainage in the tumor [49], thereby impairing drug distribution. Nanoparticles are passively delivered to the tumor through enhanced EPR effects. Nanoparticles can also be loaded with tumor-specific molecules or surface-modified peptides, antibodies, or antibody fragments for targeted aggregation in the TME. Nanoparticles can kill the tumor cells by releasing drugs or thermotherapeutic light. Grillone et al. encapsulated sorafenib and superparamagnetic iron oxide in solid lipid nanoparticles with cetyl palmitate as the lipid matrix. The nanoparticles (Sor-Mag-SLNs) had about 90% sorafenib-loading efficiency, were very stable in an aqueous environment, and improved selective targeting [50]. Zhu et al. designed resveratrol nanoparticles to inhibit RCC cell migration and invasion by regulating matrix metalloproteinase 2 (MMP-2) expression and the extracellular signal-regulated kinase (ERK) pathway [51]. Liu et al. combined tyrosine kinase inhibitors (TKI) and gold nanorods (AuNRs) with photothermal ablation in a mouse model of metastatic ccRCC. In the absence of laser irradiation, particulate or non-particulate TKI led to moderate necrosis. Irradiation with and without gold particles alone also improved tumor necrosis. However, irradiation combined with gold particles and drug-loaded nanoparticles led to complete tumor necrosis (*p <* 0.05) [52]. Kim et al. encapsulated Toll-like receptor (TLR) 7/8 agonists in PLGA nanoparticles, which were administered subcutaneously to stimulate DC activation and amplification. The nanoparticle led to the expansion of antigen-specific CD8^+^ T cells and enhanced the CTL response. The particulate agonists were superior to non-particulate agonists in preventing or treating melanoma, bladder cancer, and RCC [53]. Alsaab et al. used sorafenib in combination with tumor hypoxia-targeted nanoparticles loaded with a novel class of apoptosis inducer, CFM 4.16 (C4.16). The nanoparticle not only reversed drug resistance in RCC but also increased M1 macrophage abundance and enhanced anti-tumor cytotoxicity [54]. Chai et al. designed a tumor vaccine containing a DNA sensor (AIM2), a tumor antigen (CAIX), and a delivery system based on folic acid-grafted PEI600-CyD (H1) nanoparticles to measure the effect of immunotherapy in primary or metastatic RCC. The H1-pAIM2/pCAIX vaccine activated CAIX-specific CD8^+^ T cell proliferation and a CTL response, stimulated multifunctional CD8^+^ T cells to produce TNF-*α*, IL-2, and IFN-*γ*, and significantly inhibited tumor growth [55].

## 3. Urothelial Carcinoma

Urothelial carcinoma is a multifocal malignancy originating from the urinary epithelium, including the renal pelvis, ureter, bladder, and urethra. It is also the most common urologic tumor. The first immunotherapy modality for urothelial carcinoma, intravesical BCG instillation, received FDA approval for patients with superficial bladder tumors in 1990. Even now, BCG therapy is the standard of care for high-grade, non-invasive bladder cancer [56]. Currently, cisplatin-based combination chemotherapy is the standard of care for unresectable metastatic or advanced urothelial carcinoma; however, there are many patients with contraindications for platinum-based chemotherapy. Therefore, new treatment strategies are needed to improve the outcome of unresectable metastatic or advanced urothelial carcinoma.

### 3.1. Immunotherapy of Urothelial Carcinoma

Metastatic urothelial carcinoma is the main focus of clinical trials of immune check-point inhibitors. A single-arm phase II trial evaluated the safety and anti-tumor efficacy of pablizumab in elderly patients (≥65 and ≥75 years of age) with advanced urothelial carcinoma and ineligible for cisplatin-based chemotherapy. The results showed that the median OS was 11.3 months (95% CI 9.7–13.1 months) and PFS was 2.2 months (95% CI 2.1–3.4 months). In the high PD-L1 group, median OS was 18.5 months (95% CI 12.2–28.5 months), which was not affected by older age or poor physical status [57]. In another phase II IMvigor210 study, 220 patients received atezolizumab 1200 mg IV every 3 weeks until the clinical benefit was lost. Atezolizumab led to a median OS of 16.3 months (95% CI 10.4–24.5 months) and an ORR of 24% (95% CI 16–32%) [58]. In 2017, these findings led to the FDA approval of atezolizumab and pablizumab for patients with locally advanced or metastatic urothelial cancer who are not eligible for cisplatin-based chemotherapy. Chemotherapy in combination with immunotherapy has also raised interest in uroepithelial carcinoma. In a 1b/2 clinical trial evaluating the safety and tolerability of pabrolizumab in combination with gemcitabine and cisplatin in patients with uroepithelial carcinoma, 83 patients with T2-4AN0M0 uroepithelial carcinoma were divided into 3 groups (NCT02365766), and the trial is ongoing. There is also an ongoing clinical trial testing torexpalizumab plus gemcitabine/cisplatin in upper urinary and muscular-invasive bladder urothelial carcinoma. In a multicenter, randomized placebo-controlled phase III trial (IMvigor130), patients with untreated locally advanced or metastatic urothelial carcinoma from several countries randomly received atezolizumab plus platinum-based chemotherapy (Group A), atezolizumab monotherapy (Group B), or placebo plus platinum-based chemotherapy (Group C). Of 1213 patients, 451 (37%) were randomly assigned to group A, 362 (30%) to group B, and 400 (33%) to group C. The median follow-up for all patients was 11.8 months (IQR 6.1–17.2). At the time of the final progression-free survival analysis and interim OS analysis (31 May 2019), median OS was 16.0 months (13.9–18.9) in group A and 13.4 months (12.0–15.2) in group C (HR 0.83, 0.69–1.00; one-sided *p* = 0.027). Median OS was 15.7 months (13.1–17.8) in group B and 13.1 months (11.7–15.1) in group C (HR 1.02, 0.83–1.24). Adverse events leading to treatment discontinuation occurred in 156 (34%) patients in group A, 22 (6%) patients in group B, and 132 (34%) patients in group C. Adverse events leading to discontinuation of atezolizumab or placebo occurred in 50 (11%) patients in group A, 21 (6%) patients in group B, and 27 (7%) patients in group C. The results supported the use of atezolizumab plus platinum-based chemotherapy as a first-line treatment for metastatic urothelial carcinoma [59]. In an open-label, randomized phase II study (CheckMate-901), patients with previously untreated inoperable or metastatic urothelial cancer were randomized into four different treatment groups. In group A, patients received the combination of nabolutumab 1 mg/kg and ipilimumab 3 mg/kg four times every three weeks. Subsequently, treatment with 480 mg of nabolutumab was repeated every 4 weeks. In group B, platinum-based chemotherapy was administered every 3 weeks for 6 cycles. In group C, patients received 360 mg of nabolutumab every 3 weeks for 6 cycles in combination with gemcitabine + cisplatin chemotherapy. Subsequently, treatment with 480 mg of nabolutumab was repeated every 4 weeks. In group D, gemcitabine- and cisplatin-based chemotherapy regimens were administered as the standard treatment every 3 weeks for 6 cycles, and ORRs of 26.9% and 38.0% were achieved for the combination of 2 different doses of drugs (NCT03036098) [60]. CheckMate 032 was an open-label multi-cohort study consisting of 274 patients with unresectable locally advanced or metastatic urothelial epithelial carcinoma. The patients received nivolumab 3 mg/kg every 2 weeks as monotherapy in the first group (NIVO3). The patients received nivolumab 3 mg/kg plus Ipilimumab 1 mg/kg every 3 weeks for 4 doses, followed by nivolumab monotherapy 3 mg/kg every 2 weeks in the second group (NIVO3 + IPI1). They received nivolumab 1 mg/kg plus Ipilimumab 3 mg/kg every 3 weeks for 4 doses, followed by nivolumab 1 mg/kg plus Ipilimumab 1 mg/kg every 3 weeks for 4 doses, followed by nivolumab 1 mg/kg plus Ipilimumab 1 mg/kg every 3 weeks for 4 doses in the third group. Nivolumab monotherapy 3 mg/kg every 2 weeks (NIVO1 + IPI3) showed objective remission rates of 25.6%, 26.9%, and 38.0% in the three groups, respectively, with a median remission duration of more than 22 months in all groups [61].

### 3.2. BCG

The 2016 Edition of the EAU (European Association of Urology) guidelines for non-muscle-invasive urothelial cancer (NMIBC) recommends one year of full-dose intravesical BCG immunotherapy in intermediate-risk patients (combined with or without immediate titration). In patients with high-risk tumors, full-dose intravesical BCG therapy is recommended for 1–3 years [62]. Intravesical BCG therapy is the gold standard adjuvant treatment for NMIBC with a high risk of progression [63]. Although BCG therapy is effective, the tumor recurs in more than 40% of patients within 2 years, and approximately 10% of cases progress to muscle-invasive bladder cancer [64].

### 3.3. Active Cell Immunotherapy (ACI)

ACI focuses on isolating tumor-infiltrating cells and re-infusing these cells into patients after ex vivo proliferation. In a pilot study, tumor-draining lymph nodes were harvested for lymph node-derived T cell immunotherapy for muscle-infiltrating bladder cancer. After expansion in cell cultures, autologous tumor-specific T lymphocytes were infused, resulting in an objective response in 2 of 9 patients with no treatment-associated toxicities [65]. However, this approach needs invasive surgery and ex vivo proliferation of T cells.

### 3.4. Vaccines

S-288310 is a cancer peptide vaccine consisting of two HLA-A*24:02-restricted peptides. Thirty-eight patients with HLA-A*24:02-positive progressive urothelial cancer were enrolled in a phase I/II study. In the phase I part of the study, three patients received once-weekly subcutaneous injections of 1 mg or 2 mg of S-288310 to assess their safety and tolerability. In the phase II part of the study, 32 patients were randomized to receive either 1 mg or 2 mg of S-288310 to assess its efficacy and safety in cytotoxic T lymphocyte (CTL) induction. S-288310 was safe and well-tolerated in phase I. The objective remission rate was 6.3% and the disease control rate was 56.3% in phase II. The most common drug-related AE was injection site reactions. The median OS of patients vaccinated with S-288310 was 14.4, 9.1, and 3.7 months after 1 chemotherapy regimen, 2 regimens, and 3 or more regimens, respectively. Furthermore, 32.2% of patients survived for 2 years after first-line therapy (JapicCTI-090980) [66]. S-288310 was effective in inducing peptide-specific CTL and prolonged the survival of patients with bladder urothelial cancer. Other peptide vaccines were also tested in a single-arm, open-label phase II trial. The study included 48 patients with metastatic upper urinary tract urothelial carcinoma who received individualized peptide vaccination (PPV) after the failure of platinum-based chemotherapy. The median survival times of 7.3 months (95% CI, 5.3–13.1) and 13.0 months were reported for patients receiving combination salvage chemotherapy (95% CI, 5.7–17.5) and 4.5 months (95% CI, 1.7–10.1) for patients receiving PPV alone (*p* = 0.080). The vaccination was well-tolerated without serious adverse events [67].

### 3.5. Nanoparticles in Urothelial Carcinoma

Nanomedicine has evolved over the past few decades, showing the potential to improve early and precise diagnosis of cancers, treatment efficacy, and treatment-related adverse events.

#### 3.5.1. Diagnosis

Early detection by highly sensitive diagnostic techniques is a key factor in the management of bladder cancer. Nanotechnology has improved more efficient strategies for the diagnosis of urothelial cancer. Lin et al. combined the bladder cancer-targeting ligand, PLZ4, with a novel micellar drug delivery system. When the surface was covered with PLZ4, the targeting micelles adhered to the surface of canine bladder cancer cells and were absorbed into the target cancer cells. The micellar drug delivery system facilitates the more efficient delivery of therapeutic and imaging agents to bladder cancer xenografts [68]. Eissa et al. used gold nanoparticles in combination with lncRNA-UCA1, a long non-coding RNA upregulated in bladder cancer BC, to improve urine cytology sensitivity in low-grade and superficial bladder cancer [69]. Sweeney et al. designed a mesoporous silica core with specific scanning contrast properties and surface modifications. Using a murine bladder cancer model, they found that nanoparticles were preferentially absorbed by tumor cells more than by healthy bladder epithelium. The tumor signal increased in T1-weighted MRI and decreased in T2-weighted MRI [70]. Zhang et al. assembled antibody-modified hydroxyapatite (HAp) on a micro/nanostructured surface converted from natural shells under mild biomineralization conditions. The composition was sensitive for detecting urothelial cancer in the urine cytology of 22 patients with bladder cancer [71]. Azevedo et al. designed a magnetic nanoprobe (MNP) encapsulated by concanavalin A, wheat germ agglutinin, and *Sambucus nigra* for the selective capture of glycoproteins from the urine of patients with low-grade and high-grade non-muscle-invasive as well as muscle-infiltrating BC. They identified 63 glycoproteins that were only found in low-grade non-muscle-invasive bladder cancer, which may be useful for early diagnosis. The glycoproteins found in high-grade non-invasive and muscle-invasive tumors can also help with cancer progression monitoring [72].

#### 3.5.2. Treatment

Due to the complex anatomy of the bladder, only a small proportion of the drug can reach the target site [73,74]. In addition, the commonly used chemotherapeutic agents can impose severe side effects, such as neutropenia and infectious complications. The use of nanotechnology may improve therapeutic efficacy. Paclitaxel-loaded gelatin nanoparticles (PNP), designed by Lu et al., showed stable drug concentrations in vivo, better delivery efficacy, and low systemic toxicity [75]. A single-arm, multicenter, phase II study evaluated the efficacy and tolerability of nanoparticle albumin-bound (nab) paclitaxel (intravenous 260 mg/m^2^ every 3 weeks) in 48 patients with platinum-refractory urothelial carcinoma. Treatment continued until disease progression or unacceptable toxicity. Patients received a median of 6 cycles (range 1 to 15), and 47 patients were evaluable: 1 (2.1%) had a CR and 12 (25.5%) had a PR, with an overall remission rate of 27.7% (95% CI 17.3–44.4). The most common adverse events were fatigue (48/38, 79%), pain (48/37, 77%), alopecia (34/48, 71%), and neuropathy (30/48, 77%). The most common grade 3 or higher adverse events were pain (11/48, 23%), fatigue (5/48, 23%), hypertension (3/48, 6%), neuropathy (3/48, 6%), and joint stiffness or pain (2/48, 4%) (NCT00683059) [76]. The results showed that nab-paclitaxel was well-tolerated in patients with advanced urothelial carcinoma as a second-line treatment. Huang et al. loaded triterpenoid heteroenolide (HET), extracted from marine sponges, on biopolymeric HA (hyaluronic acid) and CHI (chitosan) nanoparticles. HET-loaded nanoparticles showed cytotoxic, anti-migratory, and pro-apoptotic effects on T24 bladder cancer cells [77]. Sahatsapan et al. designed catechol-functionalized alginate (Cat-Alg) nanoparticles combined with mangosteen transgenic extracts. The nanoparticles exhibited excellent mucosal adhesion and a cytotoxic effect on bladder cancer cells [78]. The high permeability of nanoparticles in tumors can enhance the therapeutic effect of photothermal therapy (PTT) and photodynamic therapy (PDT) in tumors. Bhandari et al. synthesized Au@TNA NPs from HAuCl4 and (TNA) tannic acid. Au@TNA NPs can carry a large number of MB (methylene blue) molecules, which can be treated by PDT and generate large amounts of reactive oxygen species (ROS) under 650 nm irradiation to effectively damage cancer cells. Interestingly, neither dark toxicity nor non-toxic PDT effects were observed in SV-HUC-1 normal bladder cell lines [79]. Zhu et al. combined bladder cancer-specific porphyrin PLZ4 nanoparticles (PNP) with PDT, to generate ROS and induce protein carbonylation and dendritic cell maturation in SV40 T/Ras double-transgenic mice with spontaneous bladder cancer. The median survival was 33.7 days in the control group compared with 44.8 days (*p* = 0.0123), 52.6 days (*p* = 0.0054), and more than 75 days (*p* = 0.0001) in the anti-PD-1, PNP PDT, and combination therapy groups, respectively [80]. Zhou et al. used macrophage-derived exosome-mimicking nanovesicles (EMVs) as a nanoplatform for delivering the CD73 inhibitor (AB680) and programmed cell death ligand 1 (PD-L1) inhibitors in a mouse model of bladder cancer. The nanocomplex (AB680@EMVs-aPDL1) enhanced tumor targeting, AB680 reduced extracellular adenosine production, and the combination therapy significantly promoted cytotoxic T lymphocyte activation and infiltration [81]. Terán-Navarro et al. combined gold glycoside particles (GNP) with 91–99 peptides of the bacterial toxin listeriolysin O (LLO) to produce a GNP-LLO91–99 nano-vaccine. The nano-vaccine blocked the attenuated immunosuppressive state of bladder cancer, increased the number of intra-tumor cytotoxic T cells and DC cells, and decreased the number of immunosuppressive T reg cells and myeloid-derived suppressor cells (MDSC) [82].

## 4. Prostate Cancer

The relatively slow progression of prostate cancer allows a more effective anti-tumor immune response. There are various tumor-associated antigens as potential targets in prostate cancer. Therefore, immunotherapy may achieve better efficacy in prostate cancer.

### 4.1. Immunotherapy for Prostate Cancer

After surgery or radiation, patients with early-stage prostate cancer are often treated with androgen deprivation therapy (ADT). Although ADT is initially effective, the tumor eventually progresses to metastatic debulking-resistant prostate cancer (mCRPC), with an estimated survival of 2–3 years [83]. At present, immunotherapy has been widely studied in the treatment of prostate cancer.

In a multi-cohort, open-label, phase II (KEYNOTE-199) study, 258 patients with metastatic castration-resistant prostate cancer (mCRPC) treated with docetaxel and hormone therapy were divided into 3 cohorts receiving pembrolizumab 200 mg every 3 weeks for up to 35 cycles. Patients in cohorts 1 and 2 had RECIST measurable PD-L1-positive and PD-L1-negative disease, respectively. Patients in cohort 3 had a bone-dominant disease, regardless of PD-L1 expression. Results showed a median OS of 9.5 months for cohort 1, 7.9 months for cohort 2, and 14.1 months for cohort 3. Treatment-related adverse events occurred in 60% of patients, which had grade 3–5 severity in 15% of cases and led to treatment discontinuation in 5% of cases (NCT02787005) [84]. Another phase III KEYNOTE-921 study of chemotherapy in combination with immunotherapy evaluated the efficacy and safety of pablizumab plus docetaxel in patients with mCRPC, previously treated with next-generation hormonal agents (NHA). The results of this study will help determine the role of pablizumab in combination with docetaxel and prednisolone or prednisolone in patients with mCRPC and provide an additional treatment option for patients who have not received chemotherapy for mCRPC and have failed or are intolerant to prior NHA therapy [85]. Nivolumab is being tested in patients with mCRPC (NCT03040791). Atezolizumab has been tested in patients with advanced stages of multiple solid tumors, such as prostate cancer (NCT02458638), and has shown anti-tumor efficacy in cervical cancer, follicular thyroid carcinoma, and thymoma [86]. The CTLA-4 antibody is also being tested in prostate cancer. In a phase I pilot trial, 14 patients with metastatic hormone-refractory prostate cancer received a single 3 mg/kg intravenous dose of Ipilimumab, and two of them showed a ≥50% decrease in prostate-specific antigen (PSA) [87]. A phase III trial, CA184–043, evaluated radiotherapy for bone metastases in men with mCRPC who had previously received docetaxel, followed by either Ipilimumab or a placebo. There was no significant improvement in OS (NCT00861614) [88], but the Ipilimumab group had higher OS rates at 2 years (25.2% vs. 16.6%), 3 years (15.3% vs. 7.9%), 4 years (10.1% vs. 3.3%), and 5 years (7.9% vs 2.7%) [89]. Combination therapies are also being tested in the CheckMate 650 trial. The combination of navulizumab 1 mg/kg with high-dose Ipilimumab (3 mg/kg) has been used for patients with mCRPC in a pre-and post-chemotherapy cohort. An ORR of 10% and a median OS of 15.2 months were achieved for patients in the post-chemotherapy cohort (NCT02985957) [90]. In the phase 1b/2 KEYNOTE-365 trial, patients with metastatic debulking-resistant prostate cancer treated with pablizumab plus doxorubicin and prednisone had a 34% decrease in PSA, an ORR (RECIST v1.1) of 23%, and a median rPFS and OS of 8.5 and 20.2 months, respectively (NCT02861573) [91]. The effect of pabrolizumab combined with prostate-specific antigen (ADXS31142), MVI-816 vaccine, or cryotherapy was measured in patients with metastatic castration-resistant prostate cancer, and OS of 16.0 months, 22.9 months, and 17.5 months were achieved, respectively [92,93,94].

### 4.2. ACI

Prostate stem cell antigen (PSCA) and prostate-specific membrane antigen (PSMA) may be new targets for CAR-T cell therapy [95,96]. Priceman et al. identified a PSCA- specific CAR with enhanced selectivity for PSCA-overexpressing tumor cells, which had acceptable efficacy in an animal model of subcutaneous prostate cancer with bone metastasis [97].

Transforming growth factor *δ* (TGF-*δ*) is highly expressed in the TME in mCRPC. Therefore, Narayan et al. developed TGF-*δ*-resistant CAR-T cells to overcome the immunosuppressive TME in mCRPC. The treatment achieved a median OS of 477 days in 13 patients [98].

V*γ*2V*δ* 2 receptors on human *γδ* T cells detect autopentadiene pyrophosphate metabolites, which are produced during isoprene biosynthesis in microbes and tumor cells. Nada et al. developed TGF-*δ*-resistant CAR-T cells by exposing V*γ*2V*δ*2 T cells to zoledronate. These cells significantly improved tumor immunotherapy, hindered tumor growth, and reduced tumor volume by 50% [99].

### 4.3. Vaccines

Sipuleucel-T is the first FDA-approved dendritic cell therapy for treating mCRPC [100]. In a phase III trial, 127 patients with asymptomatic metastatic hormone-refractory prostate cancer received 3 sipuleucel-T (*n* = 82) or placebo (*n* = 45) infusions every 2 weeks. The study reported a median survival of 25.9 months in the sipuleucel-T group and 21.4 months in the placebo group, and sipuleucel-T therapy was well-tolerated [101].

PRAME is an antigen preferentially expressed in melanoma, and PSMA is a recombinant plasmid, pPRA-PSM, encoding fragments from both antigens. Weber et al. administered MKC1106-PP, an immunotherapy regimen that co-targets PRAME, PSMA, and 2 peptides (E-PRA and E-PSM from PRAME and PSMA, respectively) for 10 patients with prostate cancer. Following this treatment, PSA decreased in four patients [102]. GX301 is a novel telomerase-based cancer vaccine consisting of four immunogenic peptides from human telomerase and two complementary adjuvants. In a randomized phase II trial, Filaci et al. administered GX301 after docetaxel chemotherapy [103]. Bilusic et al. developed a novel vaccine platform using adenovirus 5 (Ad5) vectors (E1-, E2b-), targeting three tumor-associated antigens, including PSA, short thorax, and MUC-1. Among 18 patients with metastatic CRPC, this novel vaccine led to partial remission in 1 patient and PSA decline in 5 patients, and 5 patients had stable disease for >6 months. Median progression-free survival was 22 weeks (NCT03481816) [104].

### 4.4. Nanoparticles in Prostate Cancer

Chemotherapy is widely used to control the progression of advanced prostate cancer; however, conventional chemotherapeutic agents can cause off-target toxicity and serious adverse effects. The use of therapeutic or diagnostic nanoparticles is a reliable strategy to improve the accuracy and sensitivity of diagnosis and the efficacy and specificity of medication. 

#### 4.4.1. Diagnosis

Although biopsy-proven diagnosis is currently the gold standard for prostate cancer diagnosis, specific urinary or plasma antigens and MRI are commonly used for urologic tumor screening. Nanotechnology can improve the diagnosis. PSA is a common clinical screening biomarker for prostate cancer [105]; however, because of its low sensitivity and specificity, false positives and negatives are common. Yan et al. used an electrochemically adapted sensor using graded MoS2 nanostructures and SiO_2_ nano-signal amplification for simultaneous detection of two prostate cancer biomarkers, PSA and sarcosine [106]. Yuan et al. developed PSMA-targeted GNPs to enhance GNP uptake in prostate cancer. Consistently, it was shown that prostate cancer cells had increased uptake/retention of PSMA-targeted GNPs. Furthermore, X-ray fluorescence CT (XFCT) showed the unique and non-homogeneous spatial distribution of GNPs within the tumor in vivo [107]. Zhong et al. used GoldMag nanoparticles as an MRI contrast agent coupled with the anti-epithelial cell adhesion molecule (EpCAM) DNA aptamers Eppc6 and Eppc14, which enhanced specific binding to EpCAM-positive prostate cancer cell lines and significantly reduced tumor signal intensity upon T2-weighted imaging [108]. In addition, the use of nanoparticles can help in sentinel lymph node detection to improve prostate cancer prognosis. Winter et al. used magnetometers and superparamagnetic iron oxide nanoparticles (SPIONs) to detect sentinel lymph nodes. They transrectally injected 2 mL of SPIONs into the prostate 1 day before surgery in 50 patients with prostate cancer. Lymph nodes with SPION uptake had a strongly decreased signal intensity on T2-weighted images. In addition, 890 sentinel lymph nodes were identified by SPION injection (median 17.5, interquartile range (IQR) 12–22.5). Sentinel lymph nodes were detected in all patients (100% diagnostic rate) [109].

#### 4.4.2. Treatment

Nanomaterials can improve the systemic toxicity and low efficacy of conventional chemotherapeutic agents such as paclitaxel (PTX), adriamycin, and docetaxel (DTX) in the treatment of prostate cancer. The nanomedicines for the treatment of urological cancers are summarized in Table 1.

There are numerous natural extracts with anti-prostate cancer properties. Tanaudommongkon et al. used curcumin (CUR) combined with d-*α*-tocopheryl polysuccinate 1000 (TPGS) to make CUR MT nanoparticles, which overcame CRPC cell resistance [110]. Singh et al. designed silver nanoparticles (AgNPs) of papaya leaf extract (PLE). The nanoparticle had anti-tumor potential in prostate cancer cells, DU145 [111]. Sharma et al. designed solid lipid nanoparticles (RSV-SLN) loaded with RSV (trans-resveratrol), which showed cytotoxicity against PC3 cells, prolonged somatic circulation, and an extended RSV half-life [112]. Khoobchandani et al. designed mangostin-functionalized gold nanoparticles (MGF-AuNPs), which modulated the balance between pro-tumor M2 and anti-tumor M1 macrophages, increased the expression of anti-tumor cytokines such as IL-12 and TNF-*α*, and decreased the expression of pro-tumor cytokines such as IL-10 and IL-6 in prostate cancer-bearing mice [113]. Some small-molecule drugs can also be more effective when delivered by nanoparticles.

miR-124 regulates carnitine palmitoyltransferase 1A (CPT1A) at the post-transcriptional level. Conte et al. prepared biodegradable polyethyleneimine-functionalized polyhydroxybutyrate nanoparticles (PHB-PEI NPs) to deliver miR-124 in androgen non-dependent prostate cancer PC3 cells. Transferase 1A (CPT1A) can completely inhibit the metabolism of lipid substrates by PC3 cells and prevent PC3 cell proliferation and colony formation [114]. Guo et al. used prostate-specific membrane antigen aptamer (Apt)-functionalized shell nuclei nanoparticles in paclitaxel (PTX)-resistant LNCaP (LNCaP/PTX) cells. The nanoparticles inhibited epithelial–mesenchymal transition (EMT) and re-sensitized cancer cells to PTX [115]. Wang et al. designed ionizable liposomes to carry protein kinase N3 (PKN3), which is aberrantly expressed in prostate cancer cells. The nanoparticles vigorously suppressed tumor growth (65.8%) and improved therapeutic safety [116]. Similarly, nanoparticles containing PTT/PDT have been used in prostate cancer. Akkurt et al. encapsulated indocyanine green (ICG) into polypropylene cross-ester (PLA) to produce nanoparticles, causing hyperthermia for killing PC3 cells [117]. Islam et al. used nanoparticles containing antigen-encoding mRNAs and TLR7/8 agonists to activate the CD8 T cell-mediated anti-tumor response [118]. Cole et al. used cationic RALA/pDNA nanoparticles (NPs) combined with dissolvable microneedle (MN) patches to produce a bilayer delivery system. The system effectively delivered the prostate cancer DNA vaccine to dermal and epidermal antigen-presenting cells (APC) and was able to enhance the anti-tumor immune response, delay tumor growth, and prolong survival in prostate cancer-bearing mice [119].

## 5. Conclusions and Outlook

The incidence of urologic tumors is high and still growing in many countries, which is a serious health concern. Surgery, radiotherapy, and hormone therapy are the most common treatment modalities; however, most patients experience metastasis, recurrence, or drug resistance. With the success of immunotherapy in hematologic malignancies, immunotherapy gradually entered the field of solid tumors. In this article, we focused on experimental studies of various methods of immunotherapy in the urogenital system, with a focus on ICIs. This study aimed to provide new strategies for the treatment of urologic tumors. Immunotherapy currently benefits only a small percentage of patients and causes serious irAEs; therefore, we proposed the use of nanotechnology to improve immunotherapy. The highly targeted nature of nanoparticles and their capability to contain several drugs makes early detection feasible and improves the efficacy and side effects of immunotherapy. Some nanomedicines have already entered clinical trials. In addition to the releasing site, in vivo clearance and metabolism of nanoparticles should be better understood. Furthermore, the reproducibility, cost-effectiveness, and industrialization of nanoparticles must be taken into account before clinical translation. The combination of immunotherapy and nanotechnology has been promising in the mice models of intestinal cancer, and it is worthwhile to investigate the effect of their combination in urologic tumors.

## Figures and Tables

**Figure 1 pharmaceutics-15-00546-f001:**
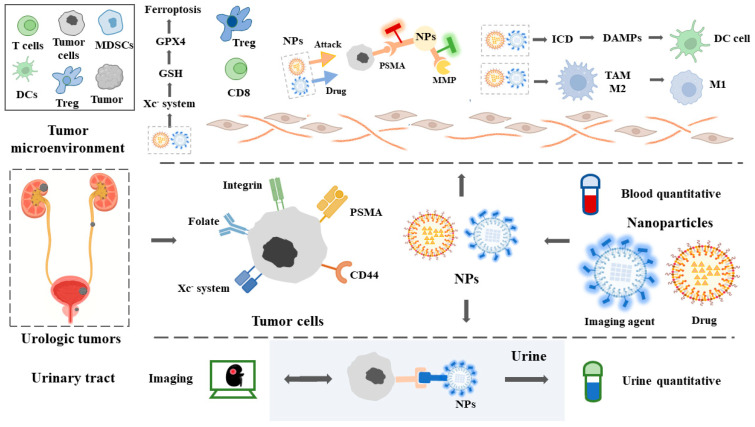
Schematic diagram of imaging, diagnosis, and therapy in urologic tumors using nanomedicine.

**Table 1 pharmaceutics-15-00546-t001:** The nanomedicines for the treatment of urological cancers.

Tumor Type	Nanoparticles	Use	References
Renal cell carcinoma	Sor-Mag-SLNs	Enhances drug delivery to tumors while reducing damage to normal tissues	[50]
	Resveratrol nanoparticles	Inhibition of RCC cell migration and invasion through regulation of MMP2 expression and the ERK pathway	[51]
	HSA-AuNR-TKI	When irradiation is paired with gold particles and drug-loaded nanoparticles, the combined therapy showed the most significant and synergistic complete tumor necrosis of 100% (*p* < 0.05)	[52]
	TLR7/8 agonists encapsulated in PLGA nanoparticles	Trigger a robust antigen-specific immune response and are highly effective as vaccine adjuvants for cancer immunotherapy	[53]
	CA IX-C4.16 NPs	Combination of CA IX-C4.16 with Sor showed targeted delivery of payload in hypoxic tumors, resulting in induction of multimodal anticancer effects, including the resurrection of apoptosis, reversal of drug resistance, and reprogramming of malfunction macrophages.	[54]
	H1-pAIM2/pCAIX	Exhibits the therapeutic efficacy of anti-renal carcinoma by enhancing tumor-specific multi-functional CD8 T cell responses	[55]
Urothelial carcinoma	Paclitaxel gelatin nanoparticles	Overcome the problem of drug dilution by newly produced urine and the sustained drug levels in tumors may decrease treatment frequency	[75]
	HA/CHI nanoparticle-aggregated HET	HA nanoparticle aggregation reinforced the cytotoxic, antimigratory, and apoptosis-inducing activities against bladder carcinoma cells and attenuated the viability–inhibitory effects on normal fibroblasts.	[77]
	Cat-Alg NPs	These NPs have the potential to be a mucoadhesive drug delivery system for bladder cancer treatment	[78]
	Au@TNA NPs	Enhance the PDT-related cytotoxicity to cancer cells, but retain a very low dark toxicity to normal cells	[79]
	PLZ4 nanoparticles	Generate ROS and induce protein carbonylation and dendritic cell maturation	[80]
	AB680@EMVs-aPDL1	Provided adequate biosafety, and enhanced tumor targeting in a mouse model of bladder cancer	[81]
	GNP-LLO_91–99_	Reduced tumor burden 4.7-fold and stimulated systemic Th1-type immune responses	[82]
Prostate cancer	CUR NPs	Restored CUR potency in both resistant DU145 and PC3 cells.	[110]
	AgNPs-PLE	Causes cell cycle arrest and apoptosis in human prostate cancer cells	[111]
	RSV-SLN	As potential carriers for drug delivery of chemotherapeutics at an extended systemic circulation and targeting efficiency at the tumor site	[112]
	MGF-AuNPs	Polarized M2-type macrophages enhance the immune response	[113]
	PHB-PEI NPs	Excellent biocompatibility and high transfection efficiency for cancer therapy	[114]
	PTX/siRNA NPs-Apt	Enhanced tumor-targeting ability and achieved superior efficacy in the subcutaneous and orthotopic PCa tumor model with minimal side effects.	[115]
	DDA-SS-DMA based delivery system encapsulating shPKN3-2459	High tumor suppression (65.8%) and treatment safety	[116]
	ICGNP	Enhance photothermal therapy	[117]
	mRNA vaccine NP	Increasing the tumor-associated antigen presentation, also promoting CD8 T cell recruitment into the tumor and enhancing the overall anti-tumor response	[118]
	RALA/pDNA NPs	Induced a tumor-specific cellular immune response, and inhibited the growth of TRAMP-C1 prostate tumors in both prophylactic and therapeutic challenge models in vivo	[119]

## Data Availability

Not applicable.

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
