# Peer review of "Nanomedicine for Combination Urologic Cancer Immunotherapy"

_pharmaceutics, 2023, doi:10.3390/pharmaceutics15020546_

Round 1
Reviewer 1 Report
Review for the article “Nanomedicine for combination urologic cancer immunotherapy”
The paper entitled “Nanomedicine for combination urologic cancer immunotherapy” described the principles of cancer immunotherapy in urology. The authors also wanted to emphasis the role of nanomendicine in immunotherapy in urologic cancer.
The objectives and the rationale of the study are clearly stated. The interpretation of results and study conclusions supported by the data. The authors clearly emphasized the strengths of their study.
The results presented in the study are very promising for clinicians.
The presentation is clear. The writing style is clear.
Reviewer 2 Report
1) The effects of chemotherapy and immunotherapy has been investigated on a wide range of tumors including urological cancers. It is strongly suggested that the authors add a new part regarding the combination of chemotherapy and immunotherapy in urological cancers using the related and recent publications
2) The similarities and differences in treatment of urological cancers with immunotherapy agents should compare with other common cancers treated with immunotherapy such as lung cancer, melanoma and so on using the relevant publications for example Alfredo Tartarone, et al - 2019 and Fausto Petrelli, et al - 2021 and Monireh Mohsenzadegan, et al - 2020 in lung cancer and so on
3) Adding a image about the content can help researcher to understand the article better.
Reviewer 3 Report
The authors provide a comprehensive review summarising current efforts to develop a targeted immuntherapy/immunodiagnosis of urological cancers via delivery with nanoparticles.
As a minor amendment, It is of interest to know whether cationic lipids - known to be cytotoxic - are involved in any of the presented nanoparticle developments. If available, such information should be added.
Round 2
Reviewer 2 Report
Accept in present form.